# Conceptualization and Validation of the Occupation Insecurity Scale (OCIS): Measuring Employees’ Occupation Insecurity Due to Automation

**DOI:** 10.3390/ijerph20032589

**Published:** 2023-01-31

**Authors:** Lara C. Roll, Hans De Witte, Hai-Jiang Wang

**Affiliations:** 1Research Group Work, Organisational & Personnel Psychology (WOPP–O2L), KU Leuven, 3000 Leuven, Belgium; hans.dewitte@kuleuven.be; 2Optentia Research Unit, Vaal Triangle Campus, North-West University, Vanderbijlpark 1900, South Africa; 3School of Management, Huazhong University of Science and Technology, Wuhan 430074, China; hjiangwang@gmail.com

**Keywords:** occupation insecurity, automation, scale validation

## Abstract

Increased use and implementation of automation, accelerated by the COVID-19 pandemic, gives rise to a new phenomenon: occupation insecurity. In this paper, we conceptualize and define occupation insecurity, as well as develop an Occupation Insecurity Scale (OCIS) to measure it. From focus groups, subject-matter expert interviews, and a quantitative pilot study, two dimensions emerged: global occupation insecurity, which refers to employees’ fear that their occupations might disappear, and content occupation insecurity, which addresses employees’ concern that (the tasks of) their occupations might significantly change due to automation. In a survey-study sampling 1373 UK employees, psychometric properties of OCIS were examined in terms of reliability, construct validity, measurement invariance (across gender, age, and occupational position), convergent and divergent validity (with job and career insecurity), external discriminant validity (with organizational future time perspective), external validity (by comparing theoretically secure vs. insecure groups), and external and incremental validity (by examining burnout and work engagement as potential outcomes of occupation insecurity). Overall, OCIS shows good results in terms of reliability and validity. Therefore, OCIS offers an avenue to measure and address occupation insecurity before it can impact employee wellbeing and organizational performance.

## 1. Introduction

The impact of automation on occupations is significant and widespread. According to a widely cited paper by Frey and Osborne [1], 47% of the total U.S. employment is at high-risk of becoming automated within the next one to two decades. Similarly, McKinsey [2] estimated that technology will replace 30% or more tasks in 60% of all jobs. In Belgium, Cedefop estimates that 40% of employees will need to acquire new skills and competencies in order to continue working or to switch to a new occupation. These predictions suggest that a large portion of the workforce will be impacted by automation in the near future.

In the context of workplace transformation, previous research has primarily focused on job insecurity, which refers to the subjective fear of losing one’s current job (quantitative job insecurity) or valued job characteristics (qualitative job insecurity), such as insurance benefits or paid leave [3,4]. In this paper, we argue that the transformations brought about by automation and digitalization refer to a broader phenomenon, which we term ‘occupation insecurity.’

In job insecurity research, a ‘job’ is defined as work one is paid for at a specific organization (Job, n.d, Merriam-Webster dictionary) [5]. A job includes a certain set of tasks and responsibilities at a specific place of work. Therefore, the term ‘job’ is context specific. In contrast, an ‘occupation’ is defined as the profession an individual has been trained in and identifies with [6]. It is a generalized term that covers jobs with similar characteristics. As such, ‘occupation’ is the umbrella term for job, employment, or business with which an individual earns money. It therefore defines one’s role in society. For example, working as an administrative employee at a specific company is a job. One can switch jobs and move to a different organization. However, as more and more administrative tasks are becoming automated, the occupation of administrative worker is increasingly disappearing, potentially forcing individuals to learn a different occupation.

Especially in the context of automation, a wider problem than individual job loss emerges: the disappearance of certain occupations, while new occupations may arise. As Frey and Osborne [1] point out, people working in certain occupations will be more likely to lose their jobs than others. In a 2021 survey conducted by PwC among 32,500 workers, 39% believed their job would become obsolete within five years, and 60% were worried that automation is putting many jobs at risk. This is one example in which reference is made to ‘jobs’ while in fact the term ‘occupation’ would be more accurate. By making occupations obsolete, those people working within those occupations are threatened by job loss. When speaking about reskilling and upskilling, the underlying meaning is for people to learn a new occupation or to expand their existing skillset to be able to work in a different occupation. Therefore, the threat of automation is wider than the threat of job loss: If someone loses a specific job, the individual can seek a new job elsewhere. If, however, the occupation is disappearing, it threatens the whole livelihood of that person and may create fear in individuals regarding whether or not they will be able to cope with the required changes. We term this phenomenon ‘occupation insecurity’ and apply the term ‘occupation’ to refer to one’s trained profession in this paper.

Occupation insecurity is brought about by labor reallocation due to automation. We therefore define it as people’s fears about the future of their occupations due to technological advancements. Therefore, occupation insecurity refers to the uncertainty about the future of one’s current profession due to newly automated processes or because other people have better technological knowledge than oneself [7,8,9].

While the world of work has always evolved (e.g., the Industrial Revolution), the pace of change is now faster than ever [10]. A major contributor has been digital transformation, creating jobs that did not exist a decade ago (e.g., app developer or cloud computing specialist), while reducing the number of jobs in other occupations (e.g., manufacturing). Not only low-skilled jobs will be lost. Algorithms for ‘Big Data’ are now rapidly reducing employees performing non-routine cognitive tasks, such as accounting and paralegal jobs [11]. Regardless of whether one takes an optimistic or pessimistic view on the future of work, the trend that whole occupations are increasingly transforming cannot be stopped, likely increasing people’s perception of occupation insecurity.

In order to be able to conduct research on this phenomenon, it is necessary that (a) the phenomenon is conceptualized and clearly defined and (b) a measurement tool is developed and empirically validated. Therefore, the aims of this present research are:Formulating a conceptualization and definition of occupation insecurity.Based on this conceptualization, developing a novel and psychometrically sound questionnaire to assess occupation insecurity called *Oc*cupation *I*nsecurity *S*cale (OCIS).Identifying the prevalence of occupation insecurity.

The OCIS scale will allow researchers to establish a nomological network of the concept of occupation insecurity and provide practitioners, as well as organizations, with an assessment tool to implement appropriate interventions to support the working population.

In terms of consequences of occupation insecurity on an individual level, related research on job insecurity suggests that the perceived threat of occupation insecurity may take a toll on one’s health, alter behaviour, and affect attitudes [12,13,14]. On an organizational level, research has demonstrated that many innovative technological implementations fail due to a lack of acceptance by employees, resulting in huge financial losses for companies every year [15]. Thus, the present research is intended to contribute to fostering successful human–computer interaction over replacement, as well as implementing effective public policy measures. An OCIS scale is required to be able to assess and examine the impact of occupation insecurity on both individual (i.e., burnout) and organizational outcomes (i.e., work engagement).

### 1.1. Occupation Insecurity in Contrast to Other Constructs

Even though there are numerous measures available to tap into different forms of insecurity-related concepts, to the best of our knowledge, there is no scale measuring the concept of occupation insecurity. One well researched concept is job insecurity, defined as ‘perceived threat of job loss and the worries related to that threat’ [3]. Job insecurity can be distinguished into the perceived threat of losing one’s job (quantitative job insecurity) and the prospect of potentially losing valued job aspects (qualitative job insecurity) [4]. In contrast to job insecurity, employees experiencing occupation insecurity perceive a much darker picture: they feel that their skills might become obsolete, and, in the near future, their occupations might not exist at all. For example, if food servers lose their jobs at a specific restaurant, they can apply for other jobs as food servers at different restaurants. However, if restaurants become more automated and much fewer food servers are needed, the food servers may be forced to learn a new occupation.

Occupation insecurity is also distinct from other existing insecurity-related concepts: Career insecurity addresses the concern that nowadays few people will have a job for life and the focus is on one’s individual career development [16]. The person may not perceive a threat to the occupation as such, but may be uncertain about how to succeed in it. More recent research on the topic has distinguished career insecurity into a larger set of eight dimensions, including career insecurity about unemployment and contractual employment conditions [17]. None of the dimensions, however, directly relate to automation.

Employment insecurity refers to the likelihood of being able to remain in paid employment in the present labor market [12]. This concept describes limitations in mobility due to the availability or lack of larger-size enterprises in a country’s economy. However, in the psychological literature, this concept is actually better captured under perceived employability, which concerns the individual’s likelihood of obtaining and retaining a job in the internal or the external labor market [18]. Thus, employability refers to individuals’ perception of whether they are able to keep or obtain a job in the present labor market. Therefore, perceived employability applies to a broader context, while occupation insecurity specifically targets automation.

Lastly, the concept of technostress has triggered a lot of research, which Tarafdar et al. [19] described as follows:

Technostress is a problem of adaptation that an individual experiences when he or she is unable to cope with, or get used to, ICTs. In the organizational context, technostress is caused by individuals’ attempts and struggles to deal with constantly evolving ICTs and the changing physical, social, and cognitive requirements related to their use. (p. 304)

Technostress is divided into several dimensions: techno-overload, techno-invasion, techno-complexity, techno-uncertainty, and techno-insecurity [19,20]. Two dimensions that appear somewhat similar to occupation insecurity could be techno-uncertainty and techno-insecurity. Items measuring ‘techno-uncertainty’ ask participants whether they work in a context of continuous technological changes and upgrades. Thus, the items are rather descriptive in terms of what is happening in the immediate surrounding rather than asking for worries related to those changes. The ‘techno-insecurity’ sub-scale consists of five items that ask for three different kinds of information: one item asks about threat to job security due to new technologies; another item asks whether constant updates of skills are required to avoid being replaced by automation, and three items ask knowledge sharing with co-workers or feeling threatened by co-workers with better technological skills. With the first item, techno-insecurity taps into perceived job insecurity, not a threat to the overarching occupation. The second item specifically refers to updating skills, and the three remaining items tap into relations with co-workers, which is not part of the hypothesized occupation insecurity construct. It can hence be concluded that techno-insecurity, though in some aspects similar to occupation insecurity, does not measure the same construct.

Taken together, all of these insecurity-related concepts do not measure whether people perceive or fear that their whole occupation is at risk of disappearing or significantly changing, which we in this article coin as ‘occupation insecurity’.

### 1.2. Characteristics of Occupation Insecurity

Parallels can be drawn between the characteristics of occupation insecurity and how job insecurity is described in the literature. Firstly, occupation insecurity involves a subjective perception [21]. The same objective situation, for example automation of the work environment, can be interpreted differently by different individuals. It can trigger insecure feelings for some, while objectively the individual would have no reason to be worried. Conversely, others may feel confident about the continuity of their profession, while the profession might be disappearing [1]. In job insecurity research, it has been found that people’s subjective perception tends to align with the objective context [22]. This is a potential characteristic of occupation insecurity that will be examined in the present research.

Secondly, occupation insecurity refers to a threat to the continued existence of people’s occupation. Insecure people might experience a discrepancy between the preferred and the experienced level of security [3]. This characteristic also contains the element of involuntariness. It is not by choice that people are faced with occupation insecurity. Individuals who voluntarily choose to leave their profession do not experience a discrepancy between the preferred and perceived state.

Thirdly, as with job insecurity, occupation insecurity is about uncertainty regarding a future situation [23]. At present, the occupation still exists. It is the anticipation of change and the uncertainty regarding what this change will bring that has an impact on the individual perception.

Fourthly, people who experience occupation insecurity are concerned about either their whole occupation disappearing, losing important features of their current occupation, or both. In a new or significantly different occupation, workers will take on different and/or new tasks and responsibilities that they will have to adapt to. Consequently, people will have to retrain or upskill to stay relevant in the future job market. This is a major difference with job insecurity, where people can look for a similar position (‘job’) elsewhere without necessarily having to switch occupations or expanding their skill set.

Based on the fourth characteristic, focus group results and the distinction between qualitative and quantitative job insecurity in the literature [4], we hypothesize that occupation insecurity consists of two sub-dimensions: global and content occupation insecurity. Global occupation insecurity is analogous to quantitative job insecurity and refers to people’s fear of their whole occupation disappearing. People who experience quantitative job insecurity are concerned about losing their job [3], while people who experience global occupation insecurity are afraid that their whole occupation may disappear. People who experience content occupation insecurity are worried that their tasks and responsibilities needed to perform their occupation may be significantly changing. This idea is derived from the concept of qualitative job insecurity, according to which individuals are concerned about losing valued job characteristics [4]. A major difference between occupation and job insecurity is that one) occupation insecurity is broader, encompassing the whole profession rather than just the individual job, and two) that occupation insecurity is specifically related to automation.

#### Previous Research Attempts

One attempt at measuring a concept similar to occupation insecurity was made by Brougham and Haar in 2018 [24]. Specifically, the authors intended to find a measure to capture “the extent to which an employee views the likelihood of Smart Technology, Artificial Intelligence, Robotics and Algorithms [STARA] impacting on their future career prospects”. Accordingly, the authors named their concept ‘STARA awareness’. However, several aspects with regards to the development of this new concept and respective measurement tool could be improved: Firstly, empirical research methods such as a literature review, expert interviews, focus groups, or similar methods could have been employed. Secondly, the STARA awareness scale is based on the job insecurity scale developed by Armstrong-Stassen [25] only. In order to ensure that the items capture all aspects of STARA awareness, it would have been useful to develop new items or base the items on various scales that measure more than the job insecurity concept. A close inspection of the items suggests that the STARA awareness scale is likely to measure job insecurity rather than STARA as a distinct construct. For example, one item reads “I am personally worried about my future in my organisation due to STARA replacing employees.” Being afraid of losing one’s job in a specific organization is defined as job insecurity [26]. Thirdly, the authors do not claim that they have adequately validated their scale. In their own words, they tested “STARA awareness to determine whether employees perceive it as a threat to their job/career” (p.240). In their paper, the authors establish reliability, but not validity, of their scale, a failure for which psychological research in general has been heavily criticized in the recent literature [27]. Overall, Brougham and Haar [24] position STARA awareness within the career planning literature, which is also a difference from the contribution occupation insecurity seeks to make.

Another measure that by virtue of its name could appear similar to occupation insecurity is the artificial intelligence anxiety scale (AIAS) [28]. AIAS consists of four dimensions: learning, sociotechnical blindness, artificial intelligence (AI) configuration, and job replacement. The first three dimensions therefore tap into aspects different from occupation insecurity: The learning dimension is comprised of questions that ask participants to rate the extent to which learning about AI creates anxiety for them. The sociocultural blindness dimension addresses the potential dangers of misuse of AI. Items of the AI configuration dimension ask participants to indicate whether they find AI intimidating or scary.

The final dimension, job replacement, consists of six items. Two of those items ask participants whether they are concerned that humanity might become too dependent on AI and lose their own reasoning skills. One item asks whether individuals are afraid that AI could make society lazier. Another item asks about the possibility of AI replacing humans. Only two items in this dimension ask whether participants are afraid that AI will take away jobs. Therefore, the dimension of job replacement within AIAS is different from OCIS for two main reasons: One, the dimension covers various different facets, only one of which deals with AI taking away jobs, and two, the emphasis is on ‘job’ replacement. The items do not cover the possibility of the whole occupation disappearing or significantly changing.

In contrast to the STARA awareness scale and AIAS, we seek to empirically validate OCIS to demonstrate its distinct properties compared to other insecurity-related concepts. Despite methodological shortcomings, Brougham and Haar’s [24] findings indicated that greater technology awareness is negatively related to organizational commitment and career satisfaction and positively related to turnover intentions, cynicism, and depression. Those preliminary findings suggest that employees are aware of technology impacting their jobs and that they feel insecure about those changes with potentially significant implications for the workplace.

### 1.3. Objectives of the Present Study

The three aims of this study relate to its eight objectives in the following ways: Aims 1 and 2, to refine the definition and to develop items covering the occupation insecurity concept and to test the psychometric properties of the OCIS scale, are addressed in Objectives 1 to 7, and Aim 3, to examine the prevalence of occupation insecurity, is tested in Objective 8. Specifically, the objectives and their respective hypotheses are as follows:

#### 1.3.1. Objective 1 (Construct Validity)

One key characteristic of occupation insecurity is people’s concern about either their occupation disappearing, the content of their current occupation significantly changing, or both. Initially, an open mind was kept to allow all kinds of possible dimensions to emerge. Based on the first results from the focus groups, the hypothesis was developed that the occupation insecurity concept would consist of two separate dimensions, namely, global and content occupation insecurity. This would be aligned with the job insecurity literature in which the job insecurity concept can be distinguished into quantitative and qualitative job insecurity [4]. Therefore, we hypothesize:

**Hypothesis 1:** 
*Occupation insecurity consists of two distinct dimensions (i.e., global and content occupation insecurity).*


#### 1.3.2. Objective 2 (Reliability)

As the next step, we examine whether the two dimensions of OCIS are reliable:

**Hypothesis 2:** *The two sub-dimensions of OCIS, i.e., global and content occupation insecurity, are reliable*.

#### 1.3.3. Objective 3 (Measurement Invariance)

To examine measurement invariance, we chose three demographic variables commonly applied to stratify a sample [29] that are also related to the scale’s future practical use in a variety of contexts and across various sample groups:

**Hypothesis 3:** *The measurement properties of the scale are invariant across various demographic groups, i.e., gender, age, and occupational position*.

#### 1.3.4. Objective 4 (Divergent and Convergent Validity)

In order to establish the validity of OCIS, it is important to examine it in relation to other insecurity concepts, first and foremost job insecurity due to their shared characteristics, and in addition career insecurity because similar to OCIS, that concept is also future-oriented:

**Hypothesis 4** : *The two dimensions of OCIS (i.e., global and content occupation insecurity) can be distinguished from the two dimensions of job insecurity (i.e., quantitative and qualitative) and career insecurity (Hypothesis 4a; divergent validity), yet OCIS will also be correlated with those constructs (Hypothesis 4b; convergent validity)*.

#### 1.3.5. Objective 5 (External Discriminant Validity)

External discriminant validity is established when OCIS has a low or null correlation with a dissimilar and distinct, yet related construct. To test this, we chose organizational future time perspective [29] because we have no theoretical reason to believe that the two concepts should overlap. At the same time, future time perspective and occupation insecurity are related in the sense that both ask participants about their anticipated occupational future. Therefore, if there is a low correlation between occupation insecurity and organizational future time perspective, external discriminant validity is established:

**Hypothesis 5:** *OCIS will have a low correlation with organizational future time perspective*.

#### 1.3.6. Objective 6 (External Validity)

In order to establish external and incremental validity, we compare the level of occupation insecurity between employees working in theoretically secure vs. insecure occupations and examine the relationship of occupation insecurity with other theoretically relevant variables (i.e., burnout and work engagement).

In order to establish which occupations are objectively considered secure and insecure, we followed the classification by Frey and Osborne [1]. For the secure group, we chose education, as this has a personal, human component that is harder to automate. For the insecure group, we selected administrative and support staff workers, who oftentimes complete repetitive tasks that can be more easily automated. By contrasting those groups, we also examine the fourth characteristic of occupation insecurity whether the subjective perception aligns with the objective context.

**Hypothesis 6:** *The objectively insecure group will perceive higher levels of occupation insecurity than the objectively secure group*.

#### 1.3.7. Objective 7 (External and Incremental Validity)

We analyze the consequences of occupation insecurity in terms of burnout and work engagement. Burnout is negative for the individual, as well as the organization, as there is the negative health impact on the employee, which may lead to reduced organizational commitment and performance and time away from work [30]. Regarding work engagement, research suggests that reduced work engagement could, for example, also negatively impact performance and organizational commitment [31].

We propose the cognitive theory of stress and coping [32] to explain how occupation insecurity might impact burnout and work engagement. This theory suggests that there are two appraisal stages. In the primary appraisal stage, individuals evaluate whether a situation is stressful, and in the secondary appraisal stage, they evaluate whether they can cope with it. If individuals perceive that the situation is stressful and that they cannot cope with it, it will lead to negative outcomes, such as higher psychological strain [33]. Applied to occupation insecurity, the stress coping theory would suggest that the evaluation of the occupation as insecure (primary appraisal) would produce perceptions of lack of control (secondary appraisal), which in turn would lead to negative outcomes, such as higher burnout and reduced work engagement. Burnout stems from continuous stress over time and has recently been defined as:

a work-related state of exhaustion that occurs among employees, which is characterized by extreme tiredness, reduced ability to regulate cognitive and emotional processes, and mental distancing. These four core dimensions of burnout are accompanied by depressed mood as well as by non-specific psychological and psychosomatic complaints. [33] (p. 4)

If individuals are afraid about the continued existence of their occupation (global occupation insecurity) or the changes that automation will bring to their professions (content occupation insecurity), they likely feel helpless, since the impact and pace of automation is to a large extent outside of their control. This, in turn, may place them at a higher risk for burnout. Therefore, we hypothesize:

**Hypothesis 7** : *Global and content occupation insecurity will be positively related to burnout*.

Furthermore, we expect that through the same mechanism, occupation insecurity will have a negative impact on work engagement. Work engagement includes three dimensions [34]:

Vigor is characterized by high levels of energy and mental resilience while working, the willingness to invest effort in one’s work, and persistence even in the face of difficulties. Dedication is characterized by a sense of significance, enthusiasm, inspiration, pride, and challenge…. The final dimension of engagement, absorption, is characterized by being fully concentrated and deeply engrossed in one’s work, whereby time passes quickly and one has difficulties with detaching oneself from work. [34] (pp. 74–75)

If individuals are negatively evaluating the situation surrounding occupation insecurity (primary appraisal) and perceive a lack of control to cope with it (secondary appraisal), the result is likely that it drains their energy (i.e., lower vigor), that they perceive less significance in what they are doing if it might soon be replaced by technology (i.e., lower dedication), and their concentration level is reduced (i.e., lower absorption). Therefore, we predict that:

**Hypothesis 8:** *Global and content occupation insecurity will be negatively related to work engagement*.

Furthermore, we examine the incremental validity of global and content occupation insecurity over quantitative and qualitative job insecurity in relation to the theoretically relevant concepts above:

**Hypothesis 9:** *After all of the variance accounted for by the two dimensions of job insecurity has been partialled out, global occupation insecurity will explain additional variance above and beyond quantitative job insecurity, and content occupation insecurity will explain additional variance above and beyond qualitative job insecurity*.

#### 1.3.8. Objective 8 (Prevalence of Occupation Insecurity)

Aligned with Aim 3, the overall prevalence of occupation insecurity in our sample is reported.

## 2. Part 1 Pilot: Conceptualization and Item Formulation of OCIS

Part 1 consists of three phases. In the first phase, we conducted focus groups. During this phase, an open mind was kept in order to allow all possible sub-dimensions of OCIS to emerge. This was aided by the fact that the Belgian focus groups were conducted by two researchers blind to the first hypothesis regarding the potential division of OCIS into two sub-dimensions. Following the focus groups, we developed a first, theory-driven set of items based on all relevant workplace insecurity scales identified from the literature. During the second phase, we conducted cognitive interviews with both subject-matter experts and members of the sample population, after which the items were refined. The third phase consisted of a pilot study to test the set of items, followed by further revision.

### 2.1. Phase 1. Focus Groups and Theory-Driven Item Generation

The purpose of phase 1 was to conceptualize occupation insecurity and develop an initial set of OCIS items. To this end, we conducted focus groups in both the UK and Belgium.

#### 2.1.1. Method

In the UK, four groups of two to four people each were interviewed in February 2019. The study was approved by the Departmental Research Ethics Committee (DREC) of the Institute of Population Ageing of the University of Oxford (UK). Out of the 11 participants total, seven were staff members of the University of Oxford, and four participants were employed by the Oxford University Press. This sample was chosen because, while education is supposed to undergo significant changes due to technological developments, experts predict that this branch will be less affected than the media industry [1]. Participants employed at the university included IT analysists and lecturers, who are generally considered less at risk of automation. At the Oxford University Press, participants worked in tax, supply chain management, and administration, all of which are generally considered more automatable. The average age was 50 years old, ranging from 24–77 years. About 88% of participants were male.

In Belgium, a total of three focus groups took place, also specifically targeting participants from occupations that are hypothesized to be more versus less susceptible to automation. Ethical approval was obtained from KU Leuven (Belgium) under file number G-2019 11 1855. The group that, according to the literature, has a high probability of disappearing consisted of administrative and white-collar workers [1,35]. This group was comprised of seven individuals: two librarians, two administrative employees, two bookkeepers, and one shop assistant. The other group of occupations with a hypothetical lower probability of being automated [35] also consisted of seven participants, of which two were engineers, two were IT specialists, two were nurses, and one was a psychologist. In addition, a third focus group was conducted with participants whose occupation had already disappeared in the 1970s, namely, with 10 ex-miners. In the Belgian focus groups, 75% of participants were male.

In addition to the focus groups, we gathered existing, empirically validated workplace insecurity scales to use as a base for theory-driven item generation.

*Procedure*. The focus groups lasted about one hour, during which participants were asked to define occupation insecurity (and its components) in contrast to other forms of workplace insecurity. Furthermore, they were asked whether they believed that people (directly or indirectly, e.g., by not recommending their occupation to others) perceive occupation insecurity. They were further requested to rate the susceptibility of their occupations to automation. They were subsequently informed about experts’ ratings and asked to comment. Research has indicated that more than 80% of all themes are discoverable within two to three focus groups [36]. Conducting between three and six focus groups is likely to uncover 90% of all emerging themes. Based on those findings, we opted to conduct a total of three and four focus groups, respectively, per country, after which it appeared that data saturation had indeed been reached.

In order to gather a comprehensive list of existing and validated workplace insecurity scales, we conducted literature searches with respective keywords and consulted experts in the field (min. five years since first publication in the research area).

#### 2.1.2. Results

From the focus groups, the following key characteristics of occupation insecurity emerged: element of uncertainty; worry/fear about the future; expected changes of tasks, and/or the whole occupation becoming obsolete. Based on the focus group interviews and those key characteristics, two sub-dimensions of occupation insecurity became evident: one for overall uncertainty regarding the future of occupations due to automation (dubbed *global occupation insecurity*) and one for uncertainty related to tasks changing due to automation (dubbed *content occupation insecurity*). Specifically, the following definitions emerged:

Global occupation insecurity: individuals’ perceived probability and/or fear of their whole occupation disappearing.

Content occupation insecurity: individuals’ perceived probability and/or fear of their occupation becoming significantly different (in terms of tasks) even if the occupation as a whole might not disappear.

In terms of existing workplace insecurity measurement tools, a total of 34 insecurity-related scales were identified (see Appendix A). Taken together, the findings from the focus groups and existing workplace insecurity scales served as a basis for the initial 26 OCIS items. In an iterative process, members of the research group, in addition to two Master’s students, generated novel items based on the focus group interviews and existing scales. Those items were subsequently discussed and further amended within the team, until all members agreed on the set of 26 items for further testing.

### 2.2. Phase 2. Cognitive Interviews

#### 2.2.1. Method

In Phase 2, the 26 initial OCIS items were refined with the help of three subject-matter experts (PhD holders and academics active in the field of work and organizational psychology) and two employees. Ethical approval was obtained from the Sub-Committee on Research Ethics and Safety of the Research Committee (ref. no. EC066/1920) of Lingnan University (Hong Kong). This sample was chosen to obtain insights into the clarity and content of the items from both experts and prospective participants.

*Procedure.* For the cognitive interviews, we applied the verbal proving technique as described by Willis [37], using both concurrent and retrospective probing. Examples of concurrent probing questions include, “What does the term ‘occupation’ mean to you?” (interpretation probe) or “How did you derive at that answer?” (general probe). Retrospective probing questions included, “Does it need more instructions or an introduction in the beginning?” and “What do you think about the answer categories?”

A detailed instruction sheet was written containing standardized questions based on the focus of this study. According to this procedure, the interviewer asks the survey question, followed by the participant’s answer. The interviewer then asks for other, specific information relevant to the question, or to the specific answer given. This technique is a combination of scripted and spontaneous probes to allow for procedural flexibility.

#### 2.2.2. Results

The interviewer made notes on the participants’ feedback and comments on each item. Results of the cognitive interviews were summarized in an Excel table and discussed within the research group. The questionnaire items were modified accordingly.

### 2.3. Phase 3. Quantitative Pilot Study

#### 2.3.1. Method

*Participants and procedure.* For the quantitative pilot survey, employees in Flanders (Belgium) were invited between autumn 2019 and spring 2020 to complete an online survey on occupation insecurity. The research was approved by the Social Ethics Committee (file no. G-2019 11 1855) of KU Leuven (Belgium). As part of the convenience sampling strategy, the survey was posted on Facebook, LinkedIn, and other social media. It was always emphasized that participation was voluntary and anonymous. In this way, 203 questionnaires were collected, of which 167 questionnaires contained sufficient data to be included in the final analysis. Specifically, only the questionnaires completed up to and including the occupation insecurity items were included, as this was the central concept in this study.

The sample (N = 167) consisted of 58.7% women. The mean age was 39.95 years (*SD* = 13.64), ranging from 19 to 68 years. The occupational level was measured using six answer options: (1) unskilled blue-collar worker, (2) skilled blue-collar worker or foremen, (3) lower level white-collar worker, (4) intermediate white-collar worker or supervisor of white-collar workers, (5) upper white-collar worker, middle management/executive staff, and (6) management or director. Afterwards, the answer options were reduced to three categories, namely, low professional level (answer options 1 and 2; 4.8%), medium professional level (answer options 3 and 4; 53.9%), and high professional level (answer options 5 and 6; 38.3%). Of all participants, 57.5% worked in the private sector, 31.1% worked for the government, and 11.4% were self-employed. Occupational categories included education (14.4%), IT sector (16.8%), administrative and bank clerks (10.2%), accounting (8.4%) healthcare (14.4%), business management (13.2%), research (5.4%), engineering (5.4%), and other categories (9%), such as military, bus assistant, butcher, site manager, and baker.

*Measures.* All 26 items of the occupation insecurity questionnaire were scored on a 5-point Likert scale from (1) “strongly disagree” to (5) “strongly agree”. The items were first developed in English and then translated to Dutch following the ‘translation–back translation’ method [38].

#### 2.3.2. Results

The items were analyzed with a principal component analysis rotated by the Varimax method with Kaiser normalization. The principal component analysis showed that occupation insecurity could be divided into two factors as expected, namely, global occupation insecurity and content job insecurity. Following this analysis, the questionnaire was reduced to 14 items. For the final selection of items, those with weak or double factor loadings were omitted (Appendix B). In addition, we selected items based on content to avoid duplication.

After this quantitative pilot, both the global and content occupation insecurity subscales were further reduced to seven items each. From separate principal component analyses on the items of the shortened scales, it was concluded that all items loaded high on their respective factors. The global occupation insecurity scale had a Cronbach’s alpha of 0.90, and the content occupation insecurity scale had one of 0.84. Both scales were thus reliable.

For global occupation insecurity, the mean was 1.77 (*SD* 0.65), and for content occupation insecurity, the mean was somewhat higher, with 2.83 (*SD* 0.68). Global and content occupation insecurity were positively correlated (*r* = 0.522, *p* < 0.01). This implies that individuals who are concerned about the survival of their occupation are also more concerned about the survival of subjectively important occupational characteristics.

Following this quantitative pilot, the items for OCIS were reassessed among the researchers one more time and amended into the final scale (see Appendix C for the final scale). For global occupation insecurity, item 6 (“I am worried that my occupation will become less significant in the future with the advancement of technology.”) was reworded into “I am worried that my occupation will *not be needed anymore* in the future due to the advancement of technology”. This change was made to make the item less ambiguous because the words “less significant” could be interpreted differently by various participants. Additionally, to ensure clarity, specific time frames were added to items 7 and 9, defining short-term as one to two years and long-term as five to ten years. Furthermore, item 12 was dropped since it was positively worded in contrast to the other remaining items and could thereby lead to confusion. The final global occupation insecurity scale thus contained six items.

For the content occupation insecurity scale, a clarifying addition was made to item 22: “I will need to perform tasks in my occupation in the future, for which I am not well trained *at the moment*”. In item 23, the word “job” was replaced with “occupational” to avoid confusion of terminology (“I am certain that my *occupational* responsibilities will change significantly due to technology before my retirement.”). Item 20 was dropped since it was the only positively worded item with the potential for confusion. Items 24 and 25 were also dropped since they had the lowest factor loadings. Instead, a new item was added to strengthen the training component of the scale: “I need additional training in technology in order to be able to continue working in my occupation.” The final content occupation insecurity scale contained five items.

## 3. Part 2 Main Study: Psychometric Properties of OCIS

In Part 2, the three aims with their respective eight objectives of this study were addressed after collecting new data.

### 3.1. Method

#### 3.1.1. Participants

The data collection agency Respondi (https://www.respondi.com/EN/) was commissioned to collect survey data from employees in the UK between 1 and 20 December 2020. The Sub-Committee on Research Ethics and Safety of the Research Committee (ref. no. EC066/1920) of Lingnan University (Hong Kong) approved this study. The goal was to spread the sample across the key demographics age, gender, and geographical region to ensure representativeness. Furthermore, Respondi was tasked with collecting an additional sample targeting a high-risk (i.e., administrative and support staff) and low-risk group (i.e., education staff) group for automation. Participants were provided with an informed consent form, in which it was emphasized that participation was voluntary, and that anonymity and confidentiality would be guaranteed. In total, 1453 complete questionnaires were collected. Response time and straightlining behaviour (i.e., participants ticked the same answer to most statements) were checked. Based on the number of questions in the survey, it was estimated that a response time under 8 min would be unreasonable. In addition, a variance across all survey items (excluding demographics) below 1 was considered straightlining behaviour. Taken together, applying these two criteria led to the removal of 80 participants.

The final sample consisted of 1373 employees, of which 165 participants belonged to the high-risk group for automation (i.e., administrative and support staff) and 277 employees belonged to the low-risk group (i.e., education staff). About 54% of participants were female. The age range was 18 to 65 years old. About 21.2% belonged to the younger group (18–34), 39.6% was in the middle-aged group (35–49), and 39.2% were in the older age group (50–65). Within the younger group, ages 18–24 years old were slightly underrepresented compared to the general population, but that is because we only sampled employees and many individuals within that age bracket are still following education. In terms of occupational position, 17% identified themselves in the low level (i.e., unskilled or skilled blue-collar worker), 63% in the medium level (i.e., lower level or intermediate white-collar worker), and 20% in the high level (i.e., upper white-collar worker or manager/director). Regarding education, 36% indicated they have a low educational level, 45% a medium level and 19% a high level.

#### 3.1.2. Measures

For the following measures, all items were rated on a 5-point scale ranging from 1 = “strongly disagree” to 5 = “strongly agree”.

Occupation insecurity was measured with the newly developed OCIS scale (see Appendix C). The final scale consists of six items for global and five items for content occupation insecurity. A sample item for global occupation insecurity is “I am worried that my occupation will not be needed anymore in the future due to the advancement of technology”. Content occupation insecurity is measured with, for example, the item: “I expect that my occupation will undergo significant changes due to technological developments.” To ensure that participants were aware of the difference between an “occupation” and a “job”, we added instructions with double-check questions to the participants before presenting the items (detailed instructions available in Appendix C). Reliability for both sub-scales was good, with Cronbach’s alpha for global occupation insecurity being 0.94 and that of content occupation insecurity being 0.83.

Quantitative job insecurity was assessed with the 4-item scale validated by Vander Elst et al. [39]. A sample item is, “Chances are, I will soon lose my job”. Reliability was very good with a Cronbach’s alpha of 0.91.

Qualitative job insecurity was measured with four items validated by Fischmann et al. [40]. An example item is “I feel insecure about the characteristics and conditions of my job in the future”. Cronbach’s alpha of 0.91 also indicated high reliability for this scale.

Career insecurity was assessed with the 4-item scale developed by Höge et al. [41], an example of which is: “It is difficult for me to plan my professional future”. Reliability was acceptable with a Cronbach’s alpha of 0.73.

Future time perspective was measured with the three-item sub-scale “focus on opportunities” of the overall scale specifically assessing future time perspective in relation to one’s occupation developed by Zacher [29]. A sample item was, “Many opportunities await me in my occupational future.” The Cronbach’s alpha of this scale was 0.91.

Burnout was measured with a preliminary version of the 12-item short Burnout Assessment Tool (BAT) scale [33]. An example item is, “At work, I feel mentally exhausted.” Reliability of this scale was high with a Cronbach’s alpha of 0.91.

Work engagement was assessed with the UWES-3 scale [31]. This scale contains one item each for vigor, dedication, and absorption. For example, the item for vigor is, “At my work, I feel bursting with energy”. This scale had a Cronbach’s alpha of 0.86.

Demographic variables measured were occupation based on the classification by Frey and Osborne [1] (1. Education, 2. Office and Administrative Support, 3. Legal, Community Service, Arts, and Media, 4. Management, Business, and Financial, 5. Computer, Engineering, and Science, 6. Healthcare Practitioners and Technical, 7. Service, 8. Sales and Related, 9. Farming, Fishing, and Forestry, 10. Construction and Extraction, 11. Installation, Maintenance, and Repair, 12. Production, 13. Transportation and Material Moving, 14. Other), gender (1 = male, 2 = female, 3 = other), age (1 = 18–24, 2 = 25–29, 3 = 30–34, 4 = 35–39, 5 = 40–44, 6 = 45–49, 7 = 50–54, 8 = 55–65), occupational position (1 = Unskilled blue-collar worker, 2 = Skilled blue-collar worker or foremen, 3 = Lower level white-collar worker, 4 = Intermediate white-collar worker or supervisor of white-collar worker, 5 = Upper white-collar worker, middle management/executive staff, 6 = Management or director), and education (1 = Primary education, 2 = Lower secondary education, 3 = Upper secondary education, 4 = Post-secondary non-tertiary education, 5 = Short-cycle tertiary education, 6 = Bachelor’s or equivalent level, 7 = Master’s or equivalent level, 8 = Doctoral or equivalent level, 9 = Other).

#### 3.1.3. Data Analysis

The analyses were performed in MPlus 8.8 and SPSS 28. In order to address Objective 1 and Hypothesis 1 that occupation insecurity consists of two distinct dimensions, factorial validity of OCIS was assessed using confirmatory factor analysis (CFA) with MLM maximum likelihood parameter estimation. Two models were tested by means of CFA. Model 1 was a one-factor model in which all items load on one general occupation insecurity factor. Model 2 adhered to our expectations that a two-factor model with global and content occupation insecurity as separate factors would fit the data better. The following goodness-of-fit-indices and respective cut-offs were used to evaluate model fit: Chi-square (χ^2^) comparative fit index (CFI) exceeding 0.90, Tucker-Lewis index (TLI) also exceeding 0.90, and the root mean square error of approximation (RMSEA) being less than or equal to 0.06 [42]. Additionally, the average variance extracted (AVE, adequate if > 0.50) was calculated [43].

Reliability (Objective 2, Hypothesis 2) was evaluated by assessing the internal consistency (Cronbach’s alpha coefficients ≥ 0.70) and composite reliability (CR, adequate, if ≥ 0.60) score of each subscale. For Objective 3 and Hypothesis 3, measurement equivalence analyses were conducted to show that the measurement properties of the scale were invariant across various demographic groups (i.e., gender, age, and occupational position). Configural invariance is the lowest level of invariance and allows one to examine whether the overall factor structure of the scale fits well for all sample groups [44]. Metric invariance is still considered weak. This indicates that each scale item similarly loads onto the specified latent factor with similar magnitude across groups. Scalar invariance is considered strong, and it tests whether item intercepts are equivalent across groups.

Convergent and divergent validity vis-à-vis (quantitative and qualitative) job and career insecurity (Objective 4, Hypotheses 4a and b) was established using CFA. To this end, two models were tested: Model 3 had all five factors loading onto one overall factor. Model 4 was a five-factor model aligned with our expectations. The same goodness-of-fit indices were used as for the first two models to establish factorial validity.

In order to demonstrate external discriminant validity by analyzing the relationship of OCIS with organizational future time perspective (Objective 5), the correlation between the two scales was examined. Additionally, it was established whether the square root of the AVE of global and content occupation insecurity, respectively, is greater than the individual correlation between those constructs and future time perspective. For Objective 6 and Hypothesis 6, *t*-tests were conducted to analyze external validity by comparing the level of occupation insecurity between employees working in theoretically secure vs. insecure occupations. External and incremental validity (Objective 7) were analyzed in the following ways by addressing the respective hypotheses: Hypothesis 7 expected a positive relationship between OCIS and burnout, and Hypothesis 8 predicted a negative relationship with work engagement. These hypotheses were analyzed with regression analyses controlling for age, gender, and occupational position (dummy-coded). Incremental validity above and beyond the variance accounted for by (quantitative and qualitative) job insecurity was analyzed using stepwise regression. In step one, the same control variables as for the previous analyses were included, followed by quantitative and qualitative job insecurity in step two. In step three, global and content occupation insecurity were added. For Objective 8, sample means and percentages were calculated to document the prevalence of occupation insecurity.

#### 3.1.4. Results

##### Construct Validity

For construct validity, we tested the hypothesis that occupation insecurity consists of the two distinct dimensions of global and content occupation insecurity (Hypothesis 1). The results from the CFA are presented in Table 1.

Model 1, in which both dimensions loaded onto one factor, did not fit the data well, with both CFI and TLI below 0.90 and RMSEA above 0.06 (see Table 2). Model 2, allowing for two factors, fits the data better, with both CFI and TLI above 0.90. However, the RMSEA value was 0.066 and was thereby just above the recommended cut-off. When inspecting the modification indices, it became apparent that allowing the error terms of the two items related to training (C4 and C5 in the final scale, see Appendix C) to correlate would improve model fit. Given that the content of these two items overlaps, an adjusted model (Model 2a) was also tested. Through this re-specification, the model fit improved with all goodness-of-fit indicators showing good results. Loadings on the global factor ranged from 0.79–0.89 and on the content factor from 0.56–0.76. Both factors were correlated 0.70. When examining the AVE, the recommended cut-off of 0.50 was exceeded for both global (AVE = 0.69) and content occupation insecurity (AVE = 0.53). Overall, Hypothesis 1 regarding the theoretically assumed distinction between global and content occupation insecurity has been confirmed.

##### Reliability

To establish reliability, Cronbach’s alpha and CR were examined. In terms of Cronbach’s alpha, both sub-dimensions of OCIS exceeded the recommended cut-off of 0.70 (global occupation insecurity = 0.94 and content occupation insecurity = 0.83). For CR, results showed that both global (CR = 0.93) and content occupation insecurity (CR = 0.85) exceeded the 0.60 cut-off value. Therefore, Hypothesis 2 was confirmed.

##### Measurement Invariance

Based on Brown [45], using the chi-square differences to determine measurement invariance is considered too conservative. Therefore, we have examined the change in CFI instead. If the change in the CFI is <0.01, then the next higher level of invariance is supported. According to this criteria, scalar invariance was supported across gender (configural: 0.960; metric: 0.959, scalar: 0.958), age (configural: 0.958; metric: 0.957; scalar: 0.948), and occupational position (configural: 0.946; metric: 0.945; scalar: 0.944). Thus, we conclude that the measurement properties of OCIS are invariant across gender, age, and occupational position, and that Hypothesis 3 is confirmed.

##### Convergent and Divergent Validity

The results for convergent validity are reported in Table 3. Model 3, in which all five factors (quantitative job insecurity, qualitative job insecurity, career insecurity, global occupation insecurity, and content occupation insecurity) load onto one factor does not fit the data well. Model 4, on the other hand, shows good model fit on all fit indices. Loadings across all factors ranged from 0.49–0.94. Correlations between factors ranged from 0.46–0.76. Therefore, Hypothesis 4a was confirmed, stating that the two dimensions of OCIS (i.e., global and content occupation insecurity) are distinct from the two dimensions of job insecurity (i.e., quantitative and qualitative) and career insecurity. Hypothesis 4b also predicted a correlation between OCIS and those constructs, which was supported by the results. (see Table 3).

##### External Discriminant Validity

Hypothesis 5 predicted that the theoretically unrelated construct of future time perspective would have a low correlation with OCIS. Indeed, in the case of global occupation insecurity, the result was insignificant (*r*(1371) = −0.02, *p* = 0.50) and, in the case of content occupation insecurity, the correlation was very small (*r*(1371) = 0.07, *p* < 0.01). To evaluate whether OCIS measures global and content occupation insecurity separately from future time perspective, the guidelines proposed by Fornell and Larcker [46] were applied. According to their criterion, discriminant validity can be demonstrated when the square root of the AVE by a construct (here global and content occupation insecurity, respectively) is greater than the correlation between the construct and the other construct under examination (here future time perspective). For global occupation insecurity, the square root of the AVE was 0.83, which exceeded the correlation with future time perspective of −0.02. Regarding content occupation insecurity, the square root of the AVE was 0.73, which also exceeded the correlation with future time perspective of 0.07. Thus, Hypothesis 5 was confirmed.

##### External Validity

In line with Hypothesis 6, the objectively insecure group of administration and support workers (global occupation insecurity: *M* = 2.35, *SD* = 0.91; content occupation insecurity: *M* = 3.14, *SD* = 0.75) compared to the objectively secure group working in education (global occupation insecurity: *M* = 1.97, *SD* = 0.87; content occupation insecurity: *M* = 2.93, *SD* = 0.88) demonstrated significantly higher levels of both global and occupation insecurity (global: *t*(440) = 4.40, *p* < 0.001; content: *t*(388) = 2.69, *p* < 0.01).

##### External and Incremental Validity

We analyzed consequences of occupation insecurity in terms of burnout and work engagement. In Hypothesis 7, we anticipated that global and content occupation insecurity would be positively related to burnout. Results are summarized in Table 4. The effects of both dimensions were significant for employee burnout after controlling for occupational position, age, and gender, and after controlling for each other (third column). Interestingly, global occupation insecurity was more strongly related to burnout than content occupation insecurity when both dimensions were simultaneously entered into the analysis. Similarly, as can be seen from Table 4, global and content occupation insecurity were significant for work engagement (Hypothesis 8). Here, content occupation insecurity was no longer significantly related to work engagement after controlling for global occupation insecurity. Thus, both Hypotheses 7 and 8 were confirmed when both dimensions were analyzed separately, as hypothesized. Global occupation insecurity, however, seemed to be more important than content occupation insecurity when analyzing burnout and work engagement.

In Hypothesis 9, we predicted that global and content occupation insecurity would explain additional variance in the relationship with burnout and work engagement after the variance accounted for by quantitative and qualitative job insecurity had been partialled out. As can be seen in Table 5, global and content occupation insecurity are able to explain additional variance in burnout in step 3 above and beyond quantitative and qualitative job insecurity included in step 2. As can be seen from the significance level, that increment in explained variance is driven by global rather than content occupation insecurity. For work engagement, neither global nor content occupation insecurity predicted additional variance above quantitative and qualitative job insecurity.

##### Prevalence of Occupation Insecurity

For Aim 3, we analyzed the prevalence of both global and occupation insecurity in the main UK sample (excluding the additional sample of high- vs. low-risk groups). This sample was mostly representative in terms of gender, region, and age. Thus, results give a tentative indication about the prevalence in the country. Since the younger generation was slightly underrepresented in the sample and our findings show that occupation insecurity tends to be higher among that age group, the results reported here might underestimate the true value of both global and content occupation insecurity.

Table 6 contains the means, the standard deviations, and percentages of the number of participants who scored lower, equal, or higher than the mid-point three on the final scale. A score of three could be considered “neither occupationally secure, nor occupationally insecure”. A score lower than three indicated “occupation security” and a score higher than three indicated “occupation insecurity”. A total of 17.2% of participants scored higher than three on global occupation insecurity. For content occupation insecurity, about 45.3% of participants selected a score higher than three. Thus, almost half of the employees were concerned about the tasks and content of their occupations significantly changing due to automation. There was, therefore, more uncertainty about changes to the occupation than about the continued existence of it as such. All but three participants who experienced global occupation insecurity (>3.00) also showed content occupation insecurity (>3.00). The reverse relationship was less straightforward: some individuals experienced content occupation insecurity (>3.00), but not global occupation insecurity (<3.00).

## 4. Discussion

In the Future of Jobs Report 2020, the World Economic Forum (WEF) shared the prediction that automation, accelerated by the COVID-19 pandemic, will significantly displace jobs. Within the next five years, the WEF expects that about 85 million jobs will be lost, while 97 million new roles may emerge. According to the report, this shift will require 50% of all employees to re- and upskill. Taken together, technological advancements and the COVID-19 pandemic are set to create a ‘double-disruption’ that is likely to transform jobs, tasks, and skills by as early as 2025 [10].

These changes appear to give rise to occupation insecurity as a new phenomenon. Results from this study are aligned with findings from job insecurity research, showing that insecurity in the workplace impacts burnout and work engagement, among other negative consequences [12,13,14]. For organizations, the implication thereof is reduced employee performance as well as failure in the implementation of new technologies. Organizations need to transform in order to stay relevant in the market, yet recent research by the Boston Consulting Group (BCG) found that only about 30% of digital transformation processes are successful, citing insecurity and reluctance of employees to adopt the new technologies as a major contributing factor [47]. In order to address employees’ worry about the future of their occupations due to new technologies, an official conceptualization as well as a valid measurement tool are required. This study set out to provide both and achieve three aims, namely, (1) to conceptualize and define occupation insecurity, (2) to develop and validate an OCIS scale to measure the phenomenon, and (3) to identify the prevalence of occupation insecurity.

### 4.1. Conceptualization of OCIS

The first aim of the study was to provide a comprehensive understanding and definition of occupation insecurity. In contrast to a ‘job’, which concerns a specific role within a certain organization, an ‘occupation’ is defined as the profession an individual has been trained in and identifies with [6]. ‘Occupation’ is the umbrella term for job, employment, or business with which an individual earns money.

In order to conceptualize occupation insecurity focus groups and cognitive interviews were conducted with both subject-matter experts and employees. From these focus groups and interviews, the following definition of occupation insecurity emerged: Occupation insecurity refers to people’s fears about the future of their occupations due to technological advancements. The study further revealed that this overarching concept of occupation insecurity can be divided into two sub-dimensions: global and content occupation insecurity.

Global occupation insecurity refers to people’s fear of their whole occupation disappearing. This type of insecurity includes worries that the individual’s entire line of work will become irrelevant and will not be needed in the future. On the other hand, content occupation insecurity addresses people’s worry that their tasks and responsibilities may be significantly changing. This type of insecurity includes concerns that certain aspects of the individual’s occupation may become automated or outsourced, leaving them with less fulfilling responsibilities, or with tasks for which they have not been adequately trained.

Overall, in line with the goals of this study, a clear and concise definition of occupation insecurity as a concept is provided. Furthermore, two key sub-dimensions are identified, namely, global and content occupation insecurity, which capture specific concerns of individuals regarding their occupational future in the context of technological advancements.

### 4.2. Development and Psychometric Evaluation of OCIS

Our second aim was to develop and validate an occupation insecurity scale (OCIS). This scale and accompanying information can be downloaded from the website www.occupationinsecurity.com. The final OCIS measure consists of 11 items, which cover the two sub-dimensions global (six items) and content (five items) occupation insecurity.

Our predictions in terms of the validity of OCIS were mostly confirmed by the results: first, OCIS showed construct validity in terms of the two distinct sub-dimensions of global and content occupation insecurity. Second, both sub-scales had good reliability. Third, measurement invariance was confirmed across age, gender, and occupational position. Fourth, convergent and divergent validity with career and (quantitative and qualitative) job insecurity was established. Fifth, external discriminant validity with future time perspective was established. Sixth, external validity was demonstrated, as the objectively more insecure participants perceived higher levels of occupation insecurity than the objectively secure group. Seventh, external and incremental validity were partially confirmed. As expected, OCIS had a significant positive relationship with burnout and a significant negative relationship with work engagement. The global, but not the content, dimension of the scale also showed incremental validity above and beyond job insecurity for burnout. For work engagement, neither global nor content occupation insecurity explained additional variance above and beyond quantitative and qualitative job insecurity. The conclusion that could be drawn from this result is that OCIS appears to add more to the negative than to the positive side. This finding is aligned with the Job Demands-Resources (JD-R) model [48]. Research has shown that demands tend to impact more on burnout than on work engagement [49]. Thus, our results can be explained by the model, though incremental validity of OCIS remains to be further examined. Overall, our findings support OCIS as a valid and reliable measure.

Our third aim was to examine the prevalence of occupation insecurity. Since our sample was mostly representative in terms of age, gender, and geographical region of the UK, tentative conclusions can be drawn regarding the prevalence of global and occupation insecurity in the country. Results showed that about 16.5% of employees experienced global occupation insecurity. For content occupation insecurity, the percentages were almost triple (46.7%). The means for both global and content occupation insecurity are comparable to the means typically found in job insecurity research [50,51,52]. Specifically, quantitative job insecurity tends to produce lower means than qualitative job insecurity. Likewise, the mean for global occupation insecurity was lower than for content occupation insecurity, further supporting the notion that more employees are impacted by content than by global occupation insecurity. Experiencing global occupation insecurity was, however, more strongly associated with impaired wellbeing than experiencing content occupation insecurity. This mirrors the assumption that quantitative job insecurity is more severe in consequences than qualitative job insecurity, as more would be lost when one becomes unemployed compared to when one becomes uncertain regarding the future of specific job characteristics [23].

### 4.3. Limitations and Suggestions for Future Research

For this study, we would like to point out the following limitations and suggestions for future research. As the data were cross-sectional, a longitudinal follow-up would be recommended in order to establish causal relationships. We aimed to achieve a sample as representative of the UK population as possible. Yet, since the younger generation was underrepresented while that age group tends to be mostly affected by occupation insecurity, the true population values are likely slightly higher than those reported here. Future efforts in gathering representative data in the UK and additional countries would be highly recommended.

Overall, the results are promising in terms of the validity of OCIS, yet this is only a first preliminary step in the validation process. More elaborate testing with additional samples is required, especially regarding the further examination of incremental validity. The scale was developed and tested in Belgium and the UK. Further validation across different countries and languages is required.

In this study, we focused on burnout and work engagement as two outcomes relevant to both the individual and the organization. The relationship of OCIS with additional variables, such as job demands and resources, and the impact of personality and job performance, remains to be examined. With additional research, the long-term goal will be to establish the nomological network of OCIS.

Since this study provides the respective scale to measure occupation insecurity, this tool can now be used to follow up on current events such as the pandemic and how increased usage of automation affects employees. OCIS opens up the possibility for screening and determining risk groups to support affected individuals and inform policy change.

Ultimately, once OCIS has been applied to evaluate the presence and extent of occupation insecurity, interventions need to be developed and empirically validated. Given that organizations need to continue to innovate and incorporate modern technology to stay relevant, preventing occupation insecurity appears unfeasible. Yet, measures can be taken to appropriately address it to prevent negative consequences such as burnout or reduced work engagement, which will benefit both the employee and the organization. Drawing on suggestions to combat job insecurity [53], four strategies could be applied, which are all designed to increase employees’ perceptions of subjective control over the situation. Research has demonstrated that experiencing control over the future of one’s employment can buffer negative stress reactions [54]. The four potential strategies are: (1) allowing workers to participate in the change process and giving them a voice, (2) increasing employability, (3) enhancing justice perceptions, and (4) increasing communication.

Regarding the first strategy, participative decision making has been shown to be a low-cost measure to increase health, job satisfaction, and reduce absenteeism in the light of job insecurity [55]. Similar positive effects were found when employees were allowed to directly participate in decision making processes through seminars and collaborative action plans [56]. For the second strategy, providing training and opportunities for skill development, the organization will benefit from the enhanced skill set of their employees, and it is a measure that has been effective in reducing the negative consequences of job insecurity [54]. In terms of the third strategy, increasing perceived fairness in the change and transformation process has produced numerous positive results, such as increased performance [57], affective commitment, satisfaction with the organization, and reduced turnover intention [58]. Lastly, it is highly relevant to inform workers regarding changes and automation-related needs as well as respective competences to acquire. Research has shown that clearly communicating future plans within an organization can effectively reduce feelings of insecurity. This can be achieved through open and timely communication, which leads to a greater sense of predictability and control for employees. Furthermore, this type of communication can also contribute to employees feeling valued and respected by management [53,59,60]. Therefore, it is recommended to research these strategies as potential interventions for occupation insecurity.

## 5. Conclusions

In this study, we have defined and conceptualized the novel phenomenon of occupation insecurity. Specifically, we defined occupation insecurity as people’s fears about the future of their occupations due to technological advancements. We further found that occupation insecurity can be divided into global and content occupation insecurity. Global occupation insecurity is defined as individuals’ perceived probability and/or fear of their whole occupation disappearing. In contrast, content occupation insecurity is defined as individuals’ perceived probability and/or fear of their occupation becoming significantly different (in terms of tasks) even if the occupation as a whole might not disappear. In a further step, we developed the OCIS scale (www.occupationinsecurity.com) to enable the measurement of occupation insecurity and provided preliminary evidence for its validity.

In order to enable workplace transformations while ensuring that employees successfully shift into their new roles, applying the OCIS scale and measuring employees’ level of occupation insecurity is a first essential step in ensuring organizational success and individual readiness for the future world of work.

## Figures and Tables

**Table 1 ijerph-20-02589-t001:** Results of the confirmatory factor analysis (CFA) of the final scale: standardized component loadings.

Scale	Item No.	Item	Estimate	S.E.
GOI	1	I am worried that my occupation will not be needed anymore in the future due to the advancement of technology.	0.885	0.007
	2	I am worried that my occupation might disappear due to automation.	0.818	0.010
	3	There is a risk that I will have to change my present occupation due to automation.	0.890	0.007
	4	I think that my occupation will not exist anymore in the future.	0.816	0.010
	5	I am afraid that I will need to switch to another occupation in the short term (1–2 years) due to technological developments.	0.788	0.011
	6	I am afraid that I will need to switch to another occupation later on in my career (5–10 years) due to technological developments.	0.872	0.008
COI	1	I expect that my occupation will undergo significant changes due to technological developments.	0.761	0.015
	2	Certain tasks of my occupation will no longer be relevant in the future.	0.730	0.016
	3	I am certain that my occupational responsibilities will change significantly due to technology before my retirement.	0.785	0.014
	4	I will need to perform tasks in my occupation in the future, for which I am not well trained at the moment.	0.557	0.012
	5	I need additional training in technology in order to be able to continue working in my occupation.	0.618	0.020

Note. GOI = global occupation insecurity; COI = content occupation insecurity; S.E. = standard error.

**Table 2 ijerph-20-02589-t002:** Model fit indices for factorial and construct validity.

	Model	χ2	df	S-Bχ^2^	CFI	TLI	RMSEA [90% CI]		Δχ^2^	*p*
1	OCIS 1-factor	1053.50	44	1.32	0.87	0.84	0.13 [0.12–0.14]			
2	OCIS 2-factor	298.36	43	1.33	0.97	0.96	0.07 [0.06–0.07]	2 vs. 1	755.14	<0.0001
2a	Adjusted OCIS 2-factor	181.57	42	1.33	0.98	0.98	0.05 [0.04–0.06]	2a vs. 1	871.93	<0.0001
3	JI-IO-CI 1-factor	5473.05	230	1.28	0.71	0.68	0.13 [0.12–0.13]			
4	JI-IO-CI 5-factor	1035.68	220	1.27	0.95	0.95	0.05 [0.05–0.06]	4 vs. 3	4437.37	<0.0001

Note. χ^2^ = chi-square, S-Bχ^2^ = Satorra-Bentler scaling factor for chi-square; df = degrees of freedom; CFI = comparative fit index; TLI = Tucker-Lewis index; RMSEA = root mean square error of approximation; Δχ^2^ = difference in chi-square; Δdf = difference in the degrees of freedom, *p* = *p*-value; JI = quantitative and qualitative job insecurity; IO = occupation insecurity; CI = career insecurity.

**Table 3 ijerph-20-02589-t003:** Correlation table of study variables to demonstrate convergent validity.

Variable	1	2	3	4
1. Global OI	-			
2. Content OI	0.61 **	-		
3. Quantitative JI	0.60 **	0.41 **	-	
4. Qualitative JI	0.61 **	0.55 **	0.69 **	-
5. CI	0.59 **	0.52 **	0.62 **	0.64 **

Note: OI = occupation insecurity; JI = job insecurity; CI = career insecurity; ** *p* < 0.01.

**Table 4 ijerph-20-02589-t004:** Linear regression results for the relationship of occupation insecurity with burnout and work engagement.

	BO			WE		
Predictors	COI	GOI	COI & GOI	COI	GOI	COI & GOI
Gender	0.07 *	0.07 **	0.07 **	0.03	0.03	0.03
Age	−0.16 **	−0.13 **	−0.13 **	0.05	0.04	0.04
Low OP	0.01	−0.02	−0.02	−0.12 **	−0.10 **	−0.10 **
High OP	−0.04	−0.01	−0.02	0.03	0.01	0.02
COI	0.27 **	/	0.09 **	−0.12 **	/	−0.02
GOI	/	0.35 **	0.30 **	/	−0.17 **	−0.15 **
R	0.34	0.41	0.41	0.18	0.22	0.22
R^2^	0.11	0.16	0.17	0.03	0.03	0.05
F-value	35.60 **	53.09 **	45.71 **	9.22 **	13.27 **	11.13 **
df	1361	1361	1361	1361	1361	1361
Change in R^2^	0.07 **	0.12 **	0.13 **	0.01 **	0.03 **	0.03 **

Note: All coefficients are standardized. Results show the second step of the linear regression. OP = occupational position; OI = occupation insecurity; BO = burnout, WE = work engagement; COI = content occupation insecurity; GOI = global occupation insecurity; df = degrees of freedom; * *p* < 0.05; ** *p* < 0.01.

**Table 5 ijerph-20-02589-t005:** Stepwise linear regression results to examine incremental validity.

	Burnout			Work Engagement		
Predictors	Step 1	Step 2	Step 3	Step 1	Step 2	Step 3
Gender	0.05	0.05 *	0.06	0.04	0.04	0.04
Age	−0.19 **	−0.17 **	−0.15	0.07 *	0.05	0.06 *
Low OP	0.02	−0.01	−0.02	0.12 **	−0.10 **	−0.10 *
High OP	−0.01	−0.02	−0.01	0.01	0.02	0.01
Quant JI		0.08 *	0.04		−0.07 *	−0.08 *
Qual JI		0.40 **	0.36 **		−0.24 **	−0.28 **
Global OI			0.12 **			0.01
Content OI			−0.02			0.06
R	0.21	0.50	0.50	0.14	0.32	0.33
R^2^	0.04	0.24	0.25	0.02	0.10	0.11
F-value	15.20 **	73.70 **	57.21 **	6.84 **	26.18 **	20.26 **
df	1361	1361	1361	1361	1361	1361
Change in R^2^	-	0.20 **	0.01 **	-	0.08 **	0.01

Note: All coefficients are standardized. OP = occupational position; JI = job insecurity; OI = occupation insecurity; df = degrees of freedom; * *p* < 0.05; ** *p* < 0.01.

**Table 6 ijerph-20-02589-t006:** Average, standard deviation, and percentage of the number of participants scoring lower, equal, and higher than three.

Scale	*M*	*SD*	Score < 3	Score = 3	Score > 3
Global occupation insecurity	2.11	0.95	78.5%	4.3%	17.2%
Content occupation insecurity	2.95	0.92	45.0%	9.7%	45.3%

Note. Score < 3 refers to low perception of occupation insecurity; score = 3 refers to neither secure nor insecure; score > 3 refers to high occupation insecurity.

## Data Availability

The data presented in this study are available upon request from the corresponding author.

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
