# Peer review of "Conceptualization and Validation of the Occupation Insecurity Scale (OCIS): Measuring Employees’ Occupation Insecurity Due to Automation"

_ijerph, 2023, doi:10.3390/ijerph20032589_

Round 1

Reviewer 1 Report

Important, interesting and timely research. Congratulations.

The peer-reviewed paper entitled "Conceptualization and Validation of the Occupation Insecurity Scale (OCIS): Measuring Employees' Occupation Insecurity Due to Automation" is, in my opinion, an important, interesting and up-to date study. It explains in detail what the study is about. The authors presented the research hypotheses in a very detailed way, and then described the methodology, research and analysis techniques in an equally detailed way. All this made the article very long, in some parts it even looks like a textbook, and in my opinion this may be the only complaint. Therefore, it may be suggested to the authors that they consider shortening it by removing repetitive phrases.

Reviewer 2 Report

The paper, although very interesting in its intentions to develop and validate an occupational insecurity scale, is difficult to read and interpret. 

In the introduction part, it is felt that too much emphasis is given to the impact of the pandemic by COVID-19 as an implementing factor for automation. We believe that since the study started in the pre-pandemic period, COVID-19 cannot be taken into consideration as the main variable.   

The materials and methods section appears to be very confusing and does not reflect an order such that the reader fully understands the work carried out by the authors. Specifically, the materials and methods blend with the objectives and results not allowing for a quick and clear understanding and reading of the work done.

The conclusions are too concise compared to the amount of work presented. 

It is felt that the work should be reshaped and/or partially divided before being resubmitted for re-evaluation

Reviewer 3 Report

Dear Authors, 

The text addresses an important issue - the fear of workers that they will be displaced from their jobs by robots and artificial intelligence (in the text: Due to Automation). In their deliberations they relied on the Theory of Work Resource Requirements, which I think is a good choice. On this occasion, it is worth pointing out to the authors that work engagement is not treated as outcomes in the theory (ditch 108-109) but as a motivational state (Baker and Demerouti, 2017, p.275).

However, my most serious doubt concerns the validity of the initial assumption that occupation insecurity has an identical structure to job insecurity. Making this assumption has the effect of narrowing down to two factors and collecting an initial pool of statements. It is regrettable that other factors were not given a chance to emerge. Job insecurity seems to me not to be a mature enough construct to base a concept on it alone. So the "theory-driven" items strategy adopted at the outset leads to confirmation of this structure in research without the opportunity to check for other factors. But there is not a good theory here from which the content of the claims can be derived. In my opinion, workers' concerns about the introduction of computers, robots and AI into the workplace raise a wide range of fears about the loss of employment or the meaning of work. I would like to draw the authors' attention to an interesting aspect that emerged from my explorations. One employee I interviewed told me that he didn't want to work with AI because he would always perform worse compared to a bot. I think that once AI is introduced into his job, he will resign on his own out of frustration of the need for positive self-esteem, even though the position will continue to exist? Unfortunately, this element is absent from the tool, as is the frustration of the need for human dominance over the robot.

specific comments on the text

In Previous Research Attempts, I missed the publication of Wang, Y.-Y., & Wang, Y.-S. (2019). Development and validation of an artificial intelligence anxiety scale: An initial application in predicting motivated learning behavior. Interactive Learning Environments, 1-16, 1-16. https://doi.org/10.1080/10494820.2019.

In hypothesis 2, a statistical criterion is given, besides, this procedure is not repeated in any other. It may be worth standardizing it.

Section 1.3.3 assumes similarity of results across groups. I have a belief that this should not be the case in all those listed, e.g., fear of failure should be higher in older workers.

Why was the concept of AVE and CR not used to see if the method factors measure separate constructs? The high correlation between the two indicates that there may be a problem here. A related doubt relates to the statements written in lines 811-818 about content occupational insecurity being contained in global occupational insecurity. This idea is in conflict with the analyses used assuming that both factors belong to the same level of generality.

1.3.5. why was the time perspective chosen? Why not another construct. A good justification is lacking here.

Verses 313-317 and 327-233 have an incomprehensible format (bullets are the wrong format).

The initial sample is too heavily male-dominated 88%. There is no information about the gender of respondents from Belgium (rows 379-385).

In rows 477-479 the statements contradict the appendix. Contrary to what is written here, several statements entered although they should not have and several conversely dropped out, although they meet the criteria of items 24 and 25 and 2 and 6 (rows 902-903).

In the section titled Reliability, I missed several analyses. Especially concerning discriminant validity: The average variance extracted has often been used to assess discriminant validity based on the following "rule of thumb": the positive square root of the AVE for each of the latent variables should be higher than the highest correlation with any other latent variable. It should be tested too.

Row 667 "If the change is < .01"-what parameter does this apply to?

In the section titled Conceptualization, my comment is about the volume of the text.  The conceptualization of the concept was highlighted in the title of the article. In the summary, it took the authors only 9 lines.

In conclusion, I consider the text worth publishing, but after making the suggested changes.

Round 2

Reviewer 2 Report

Dear authors,

We greatly appreciate your work on revising the introduction.

The materials and methods section still appears very confusing, our suggestion was to make it much more concise, representing only the materials and methods used in the study, addressing the results obtained in the discussion.

Finally, in the discussion it appears necessary to understand how the OCIS scale, as proposed by the authors, can be a useful tool to prevent job insecurity; on the contrary we believe that the scale is exclusively an evaluation tool.

Reviewer 3 Report

Thank you, I am satisfied with the clarifications obtained and the changes made to the text.

Author Response

Thank you very much, we are very pleased to hear that you are satisfied with our revisions. Many thanks again for your valuable feedback and contribution to improving our article.